# The Homestead: Developing a Conceptual Framework through Co-Creation for Innovating Long-Term Dementia Care Environments

**DOI:** 10.3390/ijerph18010057

**Published:** 2020-12-23

**Authors:** Bram de Boer, Belkis Bozdemir, Jack Jansen, Monique Hermans, Jan P. H. Hamers, Hilde Verbeek

**Affiliations:** 1Department of Health Services Research, Care and Public Health Research Institute, Maastricht University, 6229 GT Maastricht, The Netherlands; jph.hamers@maastrichtuniversity.nl; 2Living Lab in Ageing and Long-Term Care, 6229 GT Maastricht, The Netherlands; BelkisBozdemir@mgzl.nl (B.B.); JackJansen@mgzl.nl (J.J.); moniquehermans@mgzl.nl (M.H.); 3MeanderGroep Zuid Limburg, 6372 PP Landgraaf, The Netherlands

**Keywords:** long-term care, dementia, physical environment, social environment, organizational environment, environmental design

## Abstract

Alternative care environments for regular nursing homes are highly warranted to promote health and well-being of residents with dementia that are part of an age-friendly and dementia-friendly city and society. Insight is lacking on how to translate evidence-based knowledge from theory into a congruent conceptual model for innovation in current practice. This study reports on the co-creation of an alternative nursing home model in the Netherlands. A participatory research approach was used to co-create a conceptual framework with researchers, practitioners and older people following an iterative process. Results indicate that achieving positive outcomes for people with dementia, (in)formal caregivers, and the community is dependent on how well the physical, social and organizational environment are congruently designed. The theoretical underpinnings of the conceptual model have been translated into “the homestead,” which is conceptualized around three main pillars: activation, freedom and relationships. The Homestead Care Model is an illustrative example of how residential care facilities can support the development of age-friendly communities that take into consideration the needs and requirements of older citizens. However, challenges remain to implement radical changes within residential care. More research is needed into the actual implementation of the Homestead Care Model.

## 1. Introduction

The care environment plays a crucial role to support people with dementia in their daily functioning and well-being. It can be seen as an active part of care and service delivery, especially when the disease progresses and 24-hour care is required. Traditionally, nursing homes for people with dementia are confining, closed environments which are separated from communities [1]. Routines can be rigid and structured with little affordances for residents [1,2,3]. Current evidence suggests that traditional nursing homes are not effective in supporting everyday functioning. For example, nursing home residents are often inactive [4,5], display high levels of neuropsychiatric symptoms such as agitation and depression [6,7], and are restricted in their autonomy as physical and chemical restraints are often used [8,9]. As a result, a cultural change movement has been employed in nursing homes [10,11,12]. The focus is to let people with dementia live their lives as they want to, in a self-determined manner.

Alternative care environments are highly warranted to promote health and well-being of residents with dementia and are part of an age-friendly and dementia-friendly city and society. Design should facilitate human rights and dignity of people with dementia [13,14]. Radical changes in the physical, social and organizational environment are necessary to provide a care setting that enables people with dementia to be part of society. The physical design of the buildings and surroundings of innovative residential care facilities aims to follow the most recent design principles focusing on optimizing helpful stimulation, supporting movement and engagement, creating a familiar place and designing according to a clear model on the way of life at the facility [14]. An effective design of the physical environment enables people with dementia to better find their way, reduces challenging behavior such as agitation and improves independence in activities of daily living [15,16,17,18]. Furthermore, the physical environment and architectural design help to support care and activities, as well as the approach through which staff are able to maintain and reinforce the capabilities of people with dementia. Studies have suggested that higher quality of life is associated with buildings that facilitate engagement with a variety of activities both inside and outside, are familiar, provide a variety of private and community spaces, and provide opportunities to take part in domestic activities [14,19,20]. Therefore, a large variety of different types of innovative residential care facilities are developing in many countries, including the Netherlands. A recent report illustrates the variety of concepts by reporting on 84 case studies from 27 countries [21]. These include various types of small-scale homelike care models, including 24 h green care farms.

Green care farms that are 24 h have been developed as an alternative to traditional nursing homes, putting a congruent physical, social and organizational environment into practice [22]. Originating outside the healthcare sector, green care farms combine agriculture with care activities and aim to enable independence and participation in people with dementia for as long as possible. The rationale of green care farms is that people should be able to participate in daily activities as much as possible and care service delivery is integrated in daily life [23]. Furthermore, the physical environment (including animals, plants, natural elements) offers many opportunities to incorporate these activities into normal daily care practices. Residents can move more freely than in existing nursing homes as they have the opportunity to participate in outdoor, domestic, work-related, and other activities incorporated into normal daily life [22,24]. Farmers transfer their care philosophy to their staff by acting as a role model. Staff is selected based on competencies that support the individual green care farm vision and are continuously guided in providing care according to this philosophy [25,26]. Although there is large variety between green care farms (e.g., some have a degree of commercial farming such as crops, livestock, and woodland, while others do not), early evidence on outcomes for residents with dementia living at green care farms are promising. Green care farms appear to have a direct impact on the daily lives of residents. Residents living at green care farms were found to be more active, socially engaged and came outside more often [27]. This is important, as being outside was correlated with a positive mood [28]. Furthermore, the self-reported quality of life by residents was higher in green care farms compared with regular nursing homes [29]. Similar findings were found for day care services at green care farms [23,30]. Although various types of small-scale homelike care models exist, there is a lack of theoretical frameworks underpinning components and possible working mechanisms. There is an increased interest in concepts such as green houses, care villages and green care farms [10,27,31]. However, a clear theoretical framework describing the underlying principles on alternative nursing home care for people with dementia is currently lacking. Such a theoretical framework can increase our understanding of possible working mechanisms and can guide future hypothesis testing. Therefore, the novelty of the current study lies in the provision of insight into how to translate evidence-based knowledge from theory into a congruent care model.

An important facilitator of translating scientific evidence into practice is co-creation between researchers, professionals of care organizations and end-users (residents and family members) [32,33,34,35,36]. In the development of innovative dementia care settings, staff and family members have rarely reported to be involved [37]. Previous research indicates that studies that address the outcomes of co-creation processes are scarce [32]. This is problematic, given the increased focus on client-centered services within care practice, and the importance of true partnership between formal and informal care. Within care, co-creation is more than simple collaboration between stakeholders. It is the joint creation of vital goals for patients through the process of sharing knowledge and values [32,33]. This resonates with marketing concepts related to balanced centricity and stakeholder theory, which call for a situation where the interests of all actors in a network are secured [34]. Furthermore, existing collaboration initiatives within long-term care suggest that close, intensive collaboration between science and care practice is key to success of improving long-term care for older adults [35,36].

At the moment, many residential care facilities in the Netherlands are being redesigned and there is a high need for new care models. This article reports on the co-creation of an alternative nursing home model called the Homestead (in Dutch it is called de Hoeve). It presents the results of a co-creation process in which the underlying principles of green care farms have been translated into a new care model. Older people, their families and representatives, long-term care staff, management, architects and design staff have worked with researchers to put scientific knowledge on how to design the physical, social and organizational environment in everyday care into practice.

## 2. Materials and Methods

### 2.1. Design

A case study design was used studying the development of a new nursing home model called the Homestead. An existing farm-like building was acquired by the care organization, and plans were made to purposefully redesign the building in order to facilitate care provision according to the new care model, based on principles of green care farms and other innovative residential care models (e.g., putting a congruent physical, social and organizational environment in practice, and aiming to enable independence and participation in people with dementia for as long as possible). A participatory research approach was used in which researchers and practice co-created the new care model and planned for the redesign of the facility. The current study reports on a co-creation process of two years, starting August 2018 and ending in August 2020.

### 2.2. Case and Context Description

The research was conducted within a care organization in the southern part of the Netherlands. The care organization provides care across the full long-term care spectrum, including home care services, rehabilitation, palliative care and residential care in nursing homes to a wide variety of target groups. It has 17 nursing home locations, with approximately 1160 residents in total. It is a large organization with over 5000 staff members (covering home care, domestic services, nursing home care, palliative care and other staff). The care organization is part of the Living Lab in Ageing and Long-term Care, which is a formal interdisciplinary collaboration between research, education and care organizations in the southern part of the Netherlands [36].

The Homestead will be realized in a small village with approximately 5500 inhabitants. The village used to have a traditional large scale nursing home that needed to be renovated. However, the care organization chose to buy a different building that would be more fitting to implement and support the new care model.

### 2.3. Participants

The care model was developed with both practice and research in mind. Due the participatory research approach, participants of the current study included all stakeholders involved during the development of the care model (including older people, their families and representatives, long-term care staff, management, architects and design staff). The following groups were structured to guide the development process:

The core development group (*n* = 7) led the development process and managed the co-creation by deciding on the overall planning, making first drafts based on scientific literature and lessons learned in practice, and deciding on which topics to discuss in other working groups or with external experts. Furthermore, they discussed the final decisions regarding the care model. The core development group consisted of the project leader, the board of directors of the care organization, the director within the care organization, the nursing home manager, and two researchers from the university. Members of this core development group also participated in other working groups and brought information discussed in the other working groups together. Furthermore, this group was responsible for communication with the municipality (relevant civil servants and councilors) and the general practice of the region.

The project group (*n* = 13) consisted of the project leader, director within the care organization, nursing home manager, architect, client representative, janitor, care staff (both vocationally trained and higher educated nursing staff, *n* = 4), catering manager, employee from resident administration and a researcher from the university. Furthermore other (medical) staff such as medical doctors, psychologists, GPs, physiotherapists and occupational therapists gave input to the working group on relevant topics. This working group discussed everything related to the new residential care model with a specific focus on the elaboration of the most important pillars of the care model including the physical redesign of the building, possible competencies for staff and community engagement. A small portion the project group also focused on visiting many innovative care practices in the country in order to identify possible successful elements that could fit the developed care model. The following working groups were set up on specific themes to advice the project group.

Resident, family and community engagement working groups: these working groups focused on engaging residents, family and people or organizations in the community in the development of the care model. Again, these took different forms ranging from large meetings with over 50 clubs and organizations from the local community, to smaller gatherings with family members and possible future residents. Besides standard gatherings, this working group also organized an official opening of the Homestead, informing the local community about the planning and concept of the Homestead. This opening was attended by more than 500 residents from the village. This working group also informed the community by sending multiple information letters regarding the state of affairs.Staff working groups: these working groups took several different forms during the development process ranging from brainstorm sessions with large groups of care staff (e.g., all staff of the original traditional nursing home of the village), and smaller gatherings of care staff focusing on e.g., describing what a day of a resident should look like in the Homestead, how to incorporate the outside areas more, etc. Thus, these groups varied from 3 to 30 staff members (including direct care staff, registered nurses from home-care teams, social workers and case manager).Technology working group (*n* = 6): this working group consisted of a nursing home manager, an innovation manager, an information advisor, care staff, ICT staff and a researcher from the university. This working group focused on all relevant technological questions and issues in order to facilitate providing care according to the care model such as home automation, beacon technology, etc.

### 2.4. Procedure

We used common methodology in co-creation research [32,33]. The co-creation process is an iterative process of identification of needs/aims, sharing knowledge on theory and scientific evidence and balancing this with what is possible in practice. Furthermore, careful reflection on whether the practical application is in line with the intended goals was part of the process. The development of the Homestead Care Model resulted from an iterative process of determining the main pillars of the care model, incorporating recent insights from the scientific literature, having meetings and sessions within the different working groups, working visits to existing facilities and drafting concept documents describing the care model. The core development group coordinated the process and connected the different working groups by exchanging information, feeding intermediate outcomes and results and facilitated discussions. For example, if a staff working group highlighted certain design issues, members from the core development group made sure this was communicated with the architect and other members from the project planning group. Furthermore, a log was kept that showed the overall planning and kept track of all decisions made within all working groups. If decisions were made that led to certain desired actions from other stakeholders within the development process, this was shared via the log. The role of the researchers was to monitor the process, to share scientific evidence with the other stakeholders, encourage out-of-the-box thinking and highlight best-practices. Furthermore, the researchers aided in translating evidence into practice and shared critical reflections on developed ideas. Final decisions regarding the Homestead Care Model were made by the care organization.

### 2.5. Data Collection and Analyses

Due to the participatory research approach, data collection and data analysis were performed in parallel. Data were collected by the researchers through direct observation and participation of meetings related to the development of the care model. During these meetings, the researchers kept track of a log which covered the date of the meeting, the type of meeting (e.g., workshop, board meeting, staff meeting, etc.), the topic of the meeting (e.g., physical environment, staff selection, care vision), and the participants (their position/function). Furthermore, the log contained information on which decisions/actions were made based on the meetings. In addition, the researchers held separate meetings with the primary involved nursing home manager to discuss the development process. Data analysis was thematic in nature, and dependent on the stage of development, meant that in early stages of development the researchers analyzed which themes were discussed in the meetings and how scientific evidence was used to support first ideas. In later stages of development, the researchers analyzed which main pillars of the care model were identified and how these were translated into practice examples.

Furthermore, archival data was collected. All reporting that was made during the development process was shared with the research team by the involved stakeholders. This archival data contained internal documents made during meetings (e.g., minutes), preliminary care model documents, planning documents, documents created by involved external stakeholders (e.g., architect), and presentations during board meetings. The research team saved all received documents on their university server and analyzed all documents as a whole. Relevant documents were labeled regarding the topic of the document (e.g., minutes project group meeting, action/decision list project group meeting, provisional care model document). Next, documents that were given the same/similar label, where grouped together to be read and analyzed sequentially. The data were abstracted further by summarizing main themes regarding the underlying theoretical principles and practical conceptualization.

## 3. Results

First, the theoretical principles of how nursing homes ideally support a meaningful daily life of residents with dementia are highlighted. Second, these principles were translated into practice, which is detailed below.

### 3.1. Theoretical Framework

The underlying theoretical framework that has guided the co-creation process is presented in Figure 1.

Behavior and everyday functioning are the result of an interaction between the individual and her/his environment, although exact working mechanisms for people with dementia are unknown. Theories from various scientific backgrounds (i.e., psychology, nursing, gerontology, geriatrics) stress that a match is needed between the person’s needs, his/her abilities and environmental demands to elicit adequate behavior in people with dementia, also referred to as person–environment fit [3,15,17,38,39]. If environmental demands exceed a persons’ ability to cope, they will have more difficulty maintaining adequate behavior.

Within residential dementia care settings, the literature indicates three environmental components that impact everyday life and functioning for residents with dementia (see Figure 1):Physical aspects, including interior design, outdoor areas (e.g., gardens), architecture, built environment, lay-out aspects and sensory elements.Social aspects, including interactions with others in the environment. This includes resident, staff, family and friends and also the wider community and social context in which a dementia care setting is situated (e.g., local entrepreneurs, societies, and schools).Organizational aspects, including the way dementia care is organized and how the organizational culture is being perceived (e.g., values, expectations, attitudes that guide behavior of staff working in the dementia care setting).

The physical and social environment directly impact outcomes for residents with dementia. There is substantial evidence showing that the built environment can positively impact meaningful activities and quality of life for people with dementia [20]. Furthermore, restrictive physical environments can have unintended negative consequences. Therefore, the environment should contain opportunities that encourage activity [40,41,42]. Various studies show the importance of environmental aspects for people with dementia (e.g., sunlight, sounds, view, spatial layout, nature, orientation, music, privacy, autonomy, windows, comfort, facilities, staff, group size, non-institutional character, and domesticity). These studies show that careful consideration of the physical environment can support person-centered care, social interactions and daily activities of people with dementia [2,14,39,43,44,45,46,47,48]. For instance, features of the environment such as long corridors and lack of distinction between different areas may cause troubles in wayfinding [49], whereas minimizing long corridors may facilitate the domesticity of the environment [2]. Furthermore, careful consideration of signs, colors and furniture may improve orientation [49,50]. Another important aspect of the physical environment is the sensory stimulation it provides. Sensory stimulation can also be used to facilitate wayfinding and familiarity of the environment [51,52]. Environmental features such as accessibility of activity rooms, the use of artwork, plants and windows can be important factors influencing physical activity [53]. Furthermore, the use of outdoor spaces can be facilitated by incorporating environmental features such as seasonal plants and outdoor seating [54].

Although optimizing the built environment in residential aged care is important to facilitate activities and engagement, it is pivotal that changes in the physical environment are made in conjunction with considerations regarding how care should be provided [14]. Here, social aspects are crucial to consider. For instance, environmental features have been found to be able to create a familiar place for people with dementia, and to provide links to their family and community [14]. However, staff should then allow and facilitate residents to personalize their living space (e.g., by bringing photographs, paintings and pieces of furniture) [55,56]. People with dementia have to deal with the possibility of a declining ability to initiate activities and to communicate with others, showing the importance of support in maintaining social relationships. Activities can still be meaningful for people with dementia if they cause feelings of pleasure, involvement, connection and belonging. Being autonomous and having their own identity is important for people with dementia, therefore it is pivotal that staff ensure that activities are matched with roles, interests, routines and experiences of individual residents [57,58]. Over the last decade, there has been increased focus on providing person-centered care, in which the needs of people with dementia (such as the need for inclusion or identity) are considered. More recent scientific insights suggest that person-centered care insufficiently captures the interdependencies and reciprocities that underpin caring relationships, showing the importance of relationship-centered care and the reciprocity within a caring relationship [1,59,60,61,62]. Therefore, the building of relationships should be facilitated during the daily life of residents.

The theoretical model shows that the organizational context conditions any environmental design (both physical and social) and thereby influences everyday life and functioning of residents with dementia. Within long-term care, factors related to organizational culture and leadership to maintain culture are viewed as central aspects of the organizational context that impact outcomes [63,64,65]. Evidence suggests that shared values and supportive leadership for staff help in setting priorities and improve the delivery of individualized care [66,67]. For example, wards characterized by staff with positive caring climates had significantly less agitated behaviors (restlessness, wandering) compared with negative caring climate wards [66,68]. Furthermore, dementia wards characterized by an organizational culture of shared values and strong cohesion provided better quality of care [69]. The organizational environment, therefore, is an important prerequisite to innovate and directly impacts other environmental aspects. Congruence is needed between the different environmental components (physical, social and organizational), in order to promote well-being and adapted behaviors for residents with dementia and their caregivers.

### 3.2. Translation into Practice: The Homestead Care Model

The above-mentioned theoretical framework and underlying scientific principles were used in the co-creation process with all stakeholders and translated into three pillars leading to the Homestead Care Model: (1) activation, (2) freedom and (3) relationships.

The activation principle is the first pillar of the care model. Scientific evidence shows that people with dementia wish to be involved in meaningful activities and when they are more involved in these kinds of activities, this can affect their overall well-being and affect fundamental psychological needs [58,70,71,72,73]. In addition, it is known that a more active daily life is associated with aspects related to quality of life [73,74]. All stakeholders in the development process emphasized the value and importance of an active, meaningful life for people with dementia. As a result, the core development group decided that activation should be central in all facets of care delivery and in living at the Homestead in general. The project group discussed that the underlying idea is that because people with dementia can be more active during their daily life they will also remain cognitively and physically fit longer. This is in line with the “use it or lose it” principle, which suggests that if people do not use certain functionalities of the brain for a longer period of time, they will get worse at those functionalities (or lose them completely) [75,76,77]. For instance, if a person with dementia has little or no physical activity for a longer period of time, this will negatively affect his/her physical functioning (hence, he/she is less able to walk independently). The same applies for other functionalities such as communication, reasoning or emotional skills.

The freedom principle is the second pillar of the care model and refers to freedom of movement, freedom of choice and overall autonomy. Within the staff working groups, it was discussed that even though people with dementia experience limitations in daily life, their autonomy can be supported by engaging in true partnership between the residents, their family, and formal caregivers. This is also referred to as relational autonomy, which indicates that people with dementia can still exercise autonomy, regardless of being dependent on others [78,79,80,81,82]. Building on this, the core development group decided that freedom is the second important principle. This indicated that, at the Homestead, the residents should decide how they want their care to be arranged and formal caregivers can support this. When people with dementia live at home, they are in charge. As the Homestead is their new home, they are still in charge. People with dementia sometimes need help and support to express their preferences, which is something that family and formal caregivers can support with. Hence, freedom of choice is paramount in everything including living, eating, exercising, day planning, activities, and care. In the project group, examples put forward by the staff working group were mentioned. For example, the resident can choose where to eat and with whom, he/she can choose to sleep in or not, and he/she can decide to undertake an activity alone, in a group, or not at all. The environment shapes the behavior of residents and should increase personal agency of people with dementia. Agency refers to a persons’ capacity to act within his or her environment and is influenced by contextual factors such as whether there is the possibility to act or to make choices independently. This emphasizes the importance of making sure that the physical and social environment of residents encourages them to make optimal use of their living space in the most meaningful way [3,83].

The relationship principle is the third pillar of the care model. A main aspect mentioned (in particular from the resident, family and community engagement working group) was that for people with dementia, it is important that they are supported to build a bond and forge relationships with other residents and formal caregivers. Therefore, the core development decided that relationships is the third important principle. Based on the principles of relationship-centered care, care should be provided to a small group of residents, by a small, permanent team of formal caregivers. Relationship-centered care means that care goes beyond just looking at the resident (as it is in person-centered care). The focus is on optimizing the residents’ closest relationship triangle (resident–family–formal caregiver). Quality of care is defined as the perceived quality by residents (based on their expectations with regard to care and the interactions between residents, their loved ones, and their care providers). Making shared decisions about living arrangements, lifestyle, and treatment, and monitoring these decisions to improve care will result in higher experienced quality of care.

The pillars have been translated into choices in practice. Within the core development group, it was decided that all decisions made during the development of the Homestead Care Model should be made based on promoting the three pillars of the care model. This included choices regarding the physical environment (e.g., building, outdoor spaces, furnishing), as well as choices about the social and organizational environment (e.g., staffing, required competencies, involvement of the community). The consequences for the physical and social environment are mainly based on discussion within the project group, which received input from the other working groups. The practical translations are presented below.

#### 3.2.1. The Physical Environment

Within the Homestead Care Model, it is of utmost importance that residents are physically, cognitively and socially activated or can remain active themselves during their life. To implement these principles in practice, the Homestead aims to provide an optimal person–environment fit, tailoring the environment as much as possible to the needs of individual residents, family and staff [17]. The Homestead is an existing building which is a typically rural square farm, built of limestone, with a courtyard that is more than 100 m deep. It was built around the year 1400. It has a chapel attached to the courtyard. The courtyard is accessed through a large gate. The building has a rich history and is well-known by the local community (see Figure 2 for illustrative images).

The Homestead has three floors (ground, first and second), and offers a living environment for 52 people with dementia. Figure 3 shows the floor plan of the first floor. In order to respect the privacy of residents and to facilitate their autonomy in terms of self-care, there are 52 individual rooms for residents with their own sanitary facilities. Furthermore, there are communal kitchens, living rooms and activity areas. To support residents’ agency and freedom, all residents have direct access to the outdoors, regardless of where they live in the Homestead. This means that paths, access roads and places are created on all floors that support residents to go outside and seek contact in the general outdoor spaces (garden, courtyard, etc.). A portion of the outdoor space has a raised soil-level so that residents from the first floor can walk outdoors without having to take an elevator or stairs. In addition, lines of sight, walking routes, and familiarity are taken into account in the design of the Homestead to make sure residents are able to find their way independently. Orientation to find their own room, living room, or the outdoors are facilitated. All possible spaces and rooms are used to accommodate activities for residents. Hence, there will be no separate areas for nursing staff (e.g., nurses station).

Thus, the building and the outdoor spaces are inviting and inspire residents to be active. There are many options for residents to choose how they want to spend their day. Examples include activities in the garden, working with plants, taking care of animals, or just taking a walk outside. Furthermore, indoor activities such as hobbies, preparing dinner, watching a movie, playing games, or just having a pleasant chat are some of the options. Many of these activities are also organized by or for the local community, which will facilitate the building of relationships. Within the Homestead there is a restaurant for residents and family as well as for the general public. A vibrant, social gathering place for the community is created which will facilitate the social integration of residents. There is a free water tap for passing hikers or cyclists. Furthermore, playing areas for children will facilitate intergenerational interactions.

To give the residents maximum freedom of movement, there is an open door policy, meaning that no doors are locked. Residents can choose when they want to go outside. Opportunities for social interactions as well as privacy are facilitated. Furthermore, both the indoor and outdoor areas are equipped with care communication technology. By using smart technologies, a resident is supported in their freedom (e.g., camera if needed, GPS tracking, etc.). Together with residents, family, and formal caregivers it is decided which optimal degree of freedom is possible.

#### 3.2.2. The Social Environment

Within the project group one of the themes discussed was around what a typical day at the Homestead should look like, in which it was discussed how the pillars of the care model influence daily life of residents, family and staff. Practical applications of the pillars of the care model were summarized in basic principles which were described in a document called Living and Working at the Homestead. One important principle is that the Homestead is the actual house of the resident, meaning that staff are visiting people in their own house and should act accordingly. This is a crucial principle which indicates that staff should always respect residents’ privacy, and that there should always be mutual consent in all actions. This is relevant in all aspects of care delivery.

A second important principle is related to an equal partnership between residents, family and staff. People with dementia all have their own story/history, in which they create relationships with others. Their story should not stop when they move to the Homestead. Rather, residents should be able to continue their story as they see fit, and their social environment should facilitate this. Therefore, relationship centered care and social health are important principles at the Homestead [59,84]. There is active family participation and the capacity to fulfil residents’ full potential at the center of care practice. Residents should be able to manage their life, regardless of their degree of independence. At the Homestead there is no fixed day program that determines what a resident can do. Instead, together with family, formal caregivers and residents, an equal partnership is used to determine which sources should be used to meet the needs and wishes of residents. The residents are supported to make their own choices, as they would also do in their own home.

Furthermore, the transition process to a residential care facility impacts the experiences quality of care after transition. Therefore, staff at the Homestead visit the residents and family before they actually move to the Homestead in order to familiarize themselves with the home situation, preferences, needs and existing relationships of future residents. Information that is shared by residents and family before moving to the Homestead is used to optimally support the transition process and maintain the life people are used to as much as possible. Aspects that are considered are how an optimal fit can be created with the other residents, the community and how existing habits can be implemented into normal daily care practices. For example, if a resident slept until 11:00 a.m. in their home situation, this is also possible at the Homestead, or if a resident is used to participating in certain activities, they can still do that at the Homestead (even if these activities take place outside of the Homestead).

The Homestead is more than a nursing home—the social context is intensively involved and should be seen as a house of the community. It is a lively center for the neighborhood. The areas for communal activities, outdoor spaces, and the restaurant are arranged in such a way that they can also be made available to the general public. In addition, the Homestead actively seeks connection to the neighborhood by contacting specific organizations (such as the local primary school), clubs or societies in the community (e.g., local sports club, orchestra, choir, band, etc.). With these organizations they form a reciprocal relationship (focusing on what the Homestead can do for the organizations, and what the organizations can do to contribute to a meaningful life for residents). Residents are encouraged to participate as much as possible in the activities that are going on at the Homestead. There is always something to do and residents can choose what they would like to do. By including a larger social context in the Homestead, residents are still part of larger social groups and give a meaningful contribution to the community.

#### 3.2.3. The Organizational Environment

The Homestead can be seen as an innovation platform in which new scientific insights are applied in practice in the form of experiments or pilot studies. A culture is created in which learning, development and innovation, guided by scientific insights, are central to continuously improve the care that is provided to residents. Constant discussion about each other’s actions is necessary to implement the care model completely. For instance, it is important that all options for activities are not seen as separate interventions or organized activities. They are part of the normal course of events and are integrated into normal daily life at the Homestead. However, in order to successfully implement them, staff need to be reminded and think about how to do so. It is important that there is coaching staff on all levels in the organization who are constantly coaching the rest of the team. At the Homestead, this coaching is mostly performed by management and bachelor-educated nursing staff.

At the Homestead, it is important that managers facilitate innovation by creating shared values and setting priorities. Effective management is only one dimension of leadership which also involves, for example, exercising of charisma and demonstrating commitment to innovation. Being a role model for staff is an important part of implementing the care model. The style of leadership at the Homestead is therefore consultative, facilitative and flexible. Ideally, several leaders will be operating simultaneously at multiple organizational levels. Therefore, there are role models in place on all levels of the organizations (board level, director level, management level and direct care staff level). These people are given different roles within the Homestead. They are part of the development of the care model and other staff members know that they can be seen as the voice of the care model.

In line with this, important parts of the organizational environment of the Homestead are the staff roles, the competence based selection and training of staff. The three main pillars of the care model impose adjusted requirements on staff. Staff members are selected on the basis of the care model and are required to let go of traditional ways of working. Staff roles (including living counselor, well-being coach, culinary employee and manager) are formulated. For each role, the required competencies and expertise are elaborated on. These competencies include the ability to switch between the pillars of the care model in order to properly anticipate the needs of residents. Staff is required to oversee many things at the same time and they must be flexible and creative in order to integrate activities into normal daily care practices. Based on relationship centered care, involving the residents and informal caregivers as partners in the care process and the daily life of residents is one of the requested requirements. Other competencies include facilitating residents’ autonomy, focusing on remaining capabilities, being respectful and having knowledge on dementia. In order to achieve an environment that successfully implements and maintains the care model, the Homestead uses competence based selection and training for staff. All staff receive an extensive introduction and training program about the care model. The training is aimed at the pillars of the care model and covers how staff should facilitate activation, freedom, autonomy and relationship building. All staff have a trial period of six months. If it turns out that a staff member is not employing the care model correctly, he/she is not hired to work at the Homestead. Furthermore, all staff have a personal development plan which is evaluated periodically.

## 4. Discussion

This article reports on the Homestead Care Model, which is the result of a co-creation between older people, their families and representatives, long-term care staff, management, architects and design staff and researchers. Scientific knowledge shows that the physical, social and organizational environment should be congruent in order to promote overall well-being and everyday functioning of residents and caregivers in dementia care. These principles were translated into the Homestead Care Model and conceptualized around three main pillars: activation, freedom and relationships. This means that the Homestead is aimed at providing residents with an active, meaningful daily life in which they are enabled to live their lives as they want to, in a self-determined matter. Freedom of movement, freedom of choice, and overall autonomy is facilitated by focusing on remaining capabilities, rather than on limitations of residents. Furthermore, the Homestead aims to be more than a nursing home—it is a lively center for the community that facilitates relationships and enables residents to contribute and be part of society. A crucial underlying principle of the Homestead care model is that the pillars should lead to choices regarding the physical, social and organizational environment. These different facets of the environment are interrelated and achieving the aims of the Homestead Care Model is dependent on how well the physical, social and organizational environment are attuned to each other.

The Homestead Care Model is an illustrative example of how residential care facilities can support the development of age-friendly communities that take into consideration the needs and requirements of older citizens, which is an important aim of the world-wide program by the World Health Organization (WHO) on age-friendly cities and communities [85]. Furthermore, a recent age-friendly framework also highlights the importance of considering actual physical environments that are associated with age-friendly living, which was seen as a relevant addition to the WHO domains of an age-friendly city [86]. We present an example of an innovative residential care model that may facilitate a more positive social attitude towards older people and people with dementia in the community.

As previous studies have indicated, it is challenging to implement radical changes within residential care (e.g., [10]). Besides the redesign of the physical environment, a main possible barrier of successful implementation is the lack of highly qualified staff with the competences necessary to implement the Homestead Care Model. Managers are expected to apply competence-based selection for staff. Given the current workforce challenges and high turnover within residential care, this will be challenging [87]. However, the Homestead aims to facilitate a culture focusing on personal development and innovation, which is suggested to positively impact staff-turnover [88]. Staff making the transition will be trained beforehand. Furthermore, on-the-job coaching and training of this radically new way of working will be an ongoing process before and after the move.

Some limitations of the current study should be considered. First, using a participatory research approach can be considered a strength. However, such an approach requires sensitivity to the relationship between involved stakeholders [89]. In some cases, researchers needed to make deliberate decisions to trade-off their expected outcomes with what the care organization considers to be realistic outcomes. Outcomes of the current study were therefore sometimes the result of a compromise between completely evidence based principles and considerations of the care organization in terms of resources and possibilities. This however illustrates the real world application of research findings. Second, so far the Homestead Care Model is a theoretical concept which is currently being implemented in practice. More research is needed to address the gaps in knowledge regarding which actions deliver actual positive outcomes. Future studies should include outcomes both from a psychophysiological perspective (e.g., functional ability, physical fitness, blood pressure) and a psychosocial perspective (e.g., maintaining relationships, daily life, quality of life). This is in line with the WHO statement that guidance and tools are needed to support communities to make decisions which promote actions that are most likely to ensure these outcomes [90].

## 5. Conclusions

The Homestead Care Model is a translation of underlying principles that takes into consideration the interrelatedness of the physical, social and organizational environment. It incorporates scientific knowledge which indicates that the physical, social and organizational environment should be congruent in order to promote overall well-being and everyday functioning of residents and caregivers in dementia care. Future studies should focus on the actual implementation and effectiveness of the Homestead Care Model.

## Figures and Tables

**Figure 1 ijerph-18-00057-f001:**
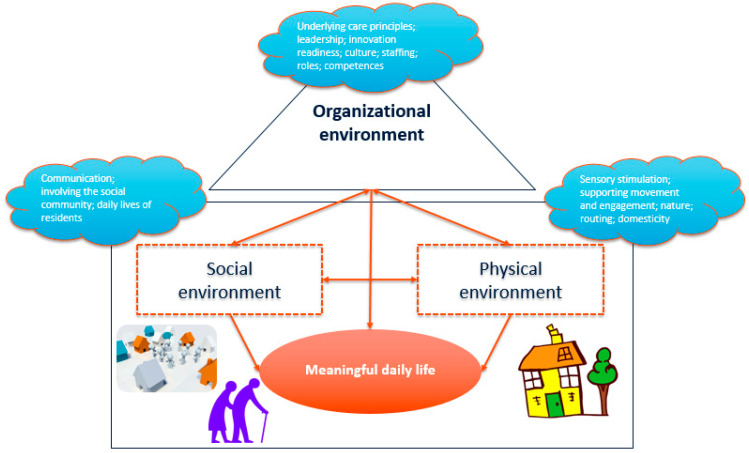
Underlying theoretical framework which integrates the physical, social and organizational environment.

**Figure 2 ijerph-18-00057-f002:**
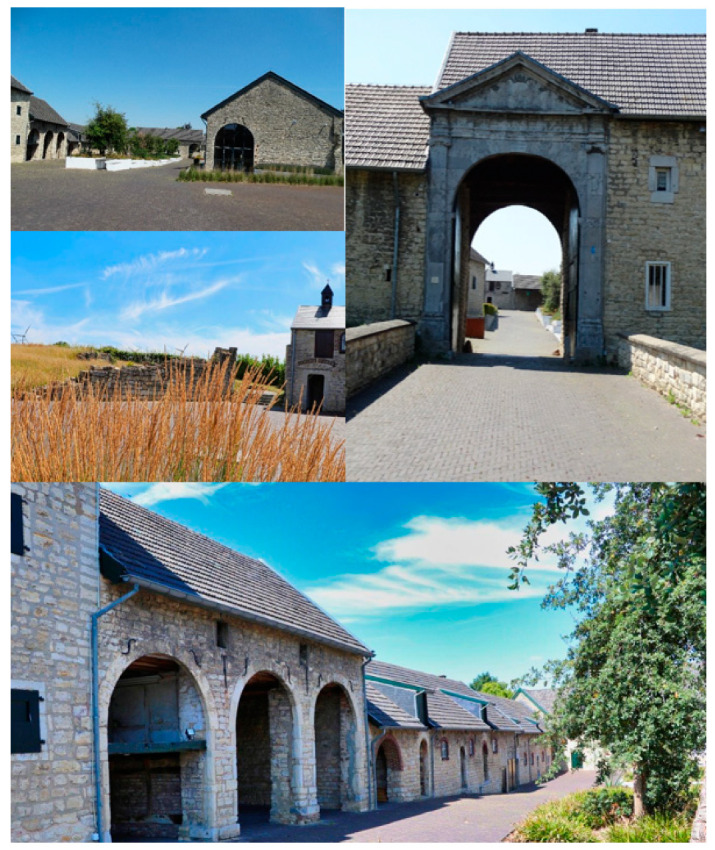
Illustrative images of the building in which the Homestead will be realized.

**Figure 3 ijerph-18-00057-f003:**
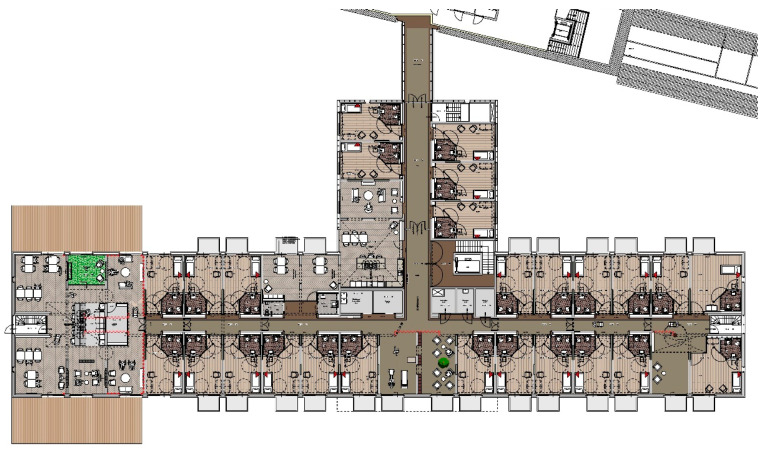
Floor plan of the Homestead (© Reproduced with permission from Widdershoven Architecten bv).

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
