# Peer review of "The Homestead: Developing a Conceptual Framework through Co-Creation for Innovating Long-Term Dementia Care Environments"

_ijerph, 2020, doi:10.3390/ijerph18010057_

Round 1
Reviewer 1 Report
Thank you for the opportunity to review this paper. While I believe the topic is an important one, I am not sure if the authors have adequately/accurately addressed the study aims according to the study title.
The authors commented that “it is currently unknown how principles of such innovative residential care models can be translated in practice”. However, in this paper the way these principles guide the project is subjective and vague.
The authors mentioned that “the co-creation process with all stakeholders and translated into three pillars that are leading in the Homestead care model: (1) activation, (2) freedom, and (3) relationships” (Lines 327-328). I don't think the authors have a clear idea of how these principles were obtained and translated. What’ more, compared with the general principles, what special are these principles under the context of the Netherlands?
The authors mentioned that “Homestead care model is dependent on how well the physical, social and organizational environment are attuned to each other”. However, the paper gives a rather sketchy description of the communication and co-creation process of different groups.
The actual positive outcomes of this case of co-creation (i.e. enhancing older people’s abilities, maintaining functional ability, or maintaining relationships) should have been assessed. The absence of this part is equivalent to the fact that the authors did an experiment but did not test the results. I think you were very honest when presenting this in the limitations of the research. But please understand that it make us question if the paper deserves to be published, as it is based on a absence of an important part of research process (i.e. the actual implementation and effectiveness of the Homestead care model).
Author Response
First of all, we would like to thank the reviewers for the valuable comments and the opportunity to revise and resubmit our manuscript. All comments are considered with care and we have implemented the suggestions in the revised manuscript. Changes and argumentation regarding the manuscript are listed below and revised text is presented in italics. When relevant, we refer to the pages and line numbers of the manuscript. Within the manuscript, track changes were used to highlight changes.
For the complete response to all reviewers please see the attached document, below you will find responses to specific comments by reviewer 1.
Reviewer 1
Before going in to detail on specific comments, we noticed that overall there appeared to be a misunderstanding about the aim of the current study. We thank the reviewer for bringing this ambiguity to our attention and apologize to have created wrong expectations about the aim of the study. It seemed as if the expectation of the reviewer was that the current study would report on implementation and outcomes of a new care model, i.e. an empirical study testing hypotheses in practice. However, the aim of the study was to develop a conceptual framework for innovation within long-term care environments, using a co-creation methodology between scientific research and care practice. The conceptual framework was developed based on values of stakeholders (e.g. older people, (in)formal caregivers, scientists, managers) and evidence from scientific research. The result of the current study is the conceptual framework and a translation of underlying principles into an example for practice.
Within current scientific literature about alternative nursing home concepts a theoretical description of underlying principles is often lacking. As also emphasized by reviewer 3, this is one of the few studies with such a focus and this highlights the novelty and utility of the manuscript. The lack of these kinds of papers impedes hypothesis testing in empirical research. Therefore, it is highly important that scientific research regarding theory development and value-creation is published. This will benefit future empirical research that focusses on implementation and effectiveness. We have made changes to the manuscript that will emphasize the added value of the study, and will prevent the abovementioned misunderstanding regarding the aim of the study.
Based on the comments of the reviewer we have made the following specific changes:
- Reviewer 1 expressed concern on the question whether we have adequately/accurately addressed the study aims according to the study title. Furthermore, the reviewer indicated that the outcomes of the co-creation should have been assessed, and that information regarding the actual implementation and effectiveness of the Homestead care model is lacking in the current manuscript.
- As said before, we thank the reviewer for bringing the ambiguity regarding the aim of the current study to our attention.
- In order to clarify the aims of the current study we have embedded this more in the introduction of the manuscript by removing a paragraph that focused too much on implementation, and adding paragraphs to better illustrate the aim of the study and the added value of co-creation. Furthermore, we have changed the abstract accordingly, and we have changed the wording of the title of the study.
- The wording of the title of the study was changed to:
- The Homestead: developing a conceptual framework through co-creation for innovating long-term dementia care environments.
- The following paragraphs were added (Page… Line….):
- The wording of the title of the study was changed to:
- In order to clarify the aims of the current study we have embedded this more in the introduction of the manuscript by removing a paragraph that focused too much on implementation, and adding paragraphs to better illustrate the aim of the study and the added value of co-creation. Furthermore, we have changed the abstract accordingly, and we have changed the wording of the title of the study.
Page 2, line 85-92:.
Although various types of small-scale homelike care models exist, there is a lack of theoretical frameworks underpinning components and possible working mechanisms. There is an increased interest in concepts such as Green Houses, care villages, or green care farms [10, 27, 31]. However, a clear theoretical framework describing the underlying principles on alternative nursing home care for people with dementia is currently lacking. Such a theoretical framework can increase our understanding of possible working mechanisms and can guide future hypothesis testing. Therefore, the novelty of the current study lies in the provision of insight into how to translate evidence-based knowledge from theory into a congruent care model.
Page 3, line 115-124:
Previous research indicates that studies that address the outcomes of co-creation processes are scarce [32]. This is problematic, given the increased focus on client-centered services within care practice, and the importance of true partnership between formal and informal care. Within care, co-creation is more than simple collaboration between stakeholders. It is the joint creation of vital goals for patients through the process of sharing knowledge and values [32-33]. This resonates with marketing concepts related to balanced centricity and stakeholder theory, which call for a situation where the interests of all actors in a network are secured [34]. Furthermore, existing collaboration initiatives within long-term care suggest that close, intensive collaboration between science and care practice is key to success of improving long-term care for older adults [35-36].
The reviewer indicated that more clarity is needed on how the general principles of the innovative residential care model were translated in practice. In addition, it was unclear how these principles within the co-creation process led to the three pillars of the Homestead care model (activation, freedom, and relationships). The reviewer suggested improving the description of the communication and co-creation of different groups involved in the process.
- In order to clarify this, we made some changes described below:
- Firstly, the procedure with regards to the how general principles were translated into practice is elaborated on in the ‘procedure’ chapter of the manuscript (page… line…), we have added information regarding the embeddedness within existing co-creation research.
Page 5, line 209-212:
We used common methodology in co-creation research [32-33]. The co-creation process is an iterative process of identification of needs/aims, sharing knowledge on theory and scientific evidence, and balancing this with what is possible in practice. Furthermore, careful reflection on whether the practical application is in line with the intended goals was part of the process.
- Secondly, in order to specify which decisions were made by whom, or when important communication took place, this is added in the manuscript. For example:
Page 8, line 346-349: All stakeholders in the development process emphasized the value and importance of an active, meaningful life for people with dementia. As a result, the core development group decided that activation should be central in all facets of care delivery and in living at the Homestead in general. The project group discussed that the underlying idea is that….
Page 8, line 358-363: Within the staff working groups, it was discussed that even though people with dementia experience limitations in daily life, their autonomy can be supported by engaging in a true partnership between the residents, their family, and formal caregivers. This is also referred to as relational autonomy, which indicates that people with dementia can still exercise autonomy, regardless of being dependent on others [78-82]. Building on this, the core development group decided that freedom is the second important principle.
Page 9, line 396-398: The consequences for the physical, social environment are mainly based on discussion within the project group, which received input from the other working groups. The practical translations are presented below.

Reviewer 2 Report
The article presents the objectives of a new nursing home project in the Netherlands for people suffering from dementia.
Authors may consider the following comments:
The work is interesting, it contains a broad justification of the actions taken. A detailed description of the project was presented. The language of the paper is comprehensive. Basically I have no major objections.
In the Keywords section, the authors included the "quality of life". How is this assessed in the current study? I am aware that the nursing home has only been functioning for a few months, so maybe it will be a prospective study comparing quality of life between standard nursing homes and Homestead? This one would be a a good summary of the actions taken by Authors, and furthermore an indication of the changes direction in the care of patients with dementia.
In some places there are missing dots or commas. i.e. line 92, Try to avoid double brackets- it may be confusing.
Author Response
First of all, we would like to thank the reviewers for the valuable comments and the opportunity to revise and resubmit our manuscript. All comments are considered with care and we have implemented the suggestions in the revised manuscript. Changes and argumentation regarding the manuscript are listed below and revised text is presented in italics. When relevant, we refer to the pages and line numbers of the manuscript. Within the manuscript, track changes were used to highlight changes.
For the complete response to all reviewers please see the attached document, below you will find responses to specific comments by reviewer 2.
Reviewer 2
We would like to thank the reviewer for the positive feedback on our justification of the actions taken, and our detailed description of the project. The reviewer made some suggestions for minor changes.
- The reviewer indicated that ‘quality of life’ was one of the keywords used. However, it was unclear how this is assessed in the current study, or how this will be assessed in future studies regarding the Homestead care concept.
- We thank the reviewer for bringing this to our attention and we have deleted this keyword
- The reviewer indicated that there were some missing dots or commas, and that double brackets should be avoided
- We have checked the manuscript multiple times and have removed all double brackets from the manuscript. Furthermore, the issue of missing dots and commas has been addressed accordingly.

Reviewer 3 Report
Summary: The current manuscript is aimed to reports on the co-creation and redesign of an alternative nursing home model in the Netherlands. Results are based on the underlying scientific hypothesis that achieving positive outcomes for people with dementia, (in)formal caregivers, and the community is dependent on how well the physical, social and organizational environment are congruently designed.
I think the study is more interesting, and I have only a few minor comments.
Specific comments follow:
Introduction: The introduction fits with the goal of the study.
Methods: Some specific information should be added.
Results and discussion: The summary of the study better explain the plan of the study. Moreover, in order to improve the results, it is useful to analyze some psychophysiological aspect that could be have a role in cognitive impairment, such as blood pressure, hrv and bmi.
For a review, see:
Forte, G., & Casagrande, M. (2019). Heart rate variability and cognitive function: a systematic review. Frontiers in neuroscience, 13, 710.
Favieri, F., & Casagrande, M. (2019). The executive functions in overweight and obesity: a systematic review of neuropsychological cross-sectional and longitudinal studies. Frontiers in Psychology, 10, 2126.
Forte, G., De Pascalis, V., Favieri, F., & Casagrande, M. (2020). Effects of Blood Pressure on Cognitive Performance: A Systematic Review. Journal of Clinical Medicine, 9(1), 34.
Forte, G.; Casagrande, M. Effects of Blood Pressure on Cognitive Performance in Aging: A Systematic Review. Brain Sci. 2020, 10, 919.
General comment: Generally, I found the study really interesting, although novelty and utility were few emphasized. My advice to the authors is to highlight the innovative nature of the study. This is one of the few study on these fields, and this aspect could be emphasized.
Author Response
First of all, we would like to thank the reviewers for the valuable comments and the opportunity to revise and resubmit our manuscript. All comments are considered with care and we have implemented the suggestions in the revised manuscript. Changes and argumentation regarding the manuscript are listed below and revised text is presented in italics. When relevant, we refer to the pages and line numbers of the manuscript. Within the manuscript, track changes were used to highlight changes.
For the complete response to all reviewers please see the attached document, below you will find responses to specific comments by reviewer 3.
Reviewer 3
We would like to thank the reviewer for the positive feedback on our manuscript. We are pleased to see that the reviewer found the study interesting and appreciate the comments made based on the innovative nature, the novelty and utility of our study. The reviewer made some suggestions for minor changes.
- Reviewer 3 indicated that in order to improve the results and discussion, it is useful to analyze some psychophysiological aspect that can have a role in cognitive impairment, such as blood pressure or BMI.
- Although the current study was aimed at conceptually presenting a new care model as a result of close collaboration between involved stakeholders. Next steps include scientific studies regarding the effectiveness of the care model. In these future studies psychophysiological aspects will be part of possible outcome measures. We have added this in the discussion section of the manuscript:
- Future studies should include outcomes both from a psychophysiological perspective (e.g. functional ability, physical fitness, blood pressure) and a psychosocial perspective (e.g. maintaining relationships, daily life, quality of life)
- The reviewer expressed that this is one of the few studies in the field, and that the innovative nature, novelty and utility deserves more emphasis.
We have incorporated this more in the introduction of the manuscript, in which we emphasize the lack of theoretical grounds of alternative nursing home concepts. Furthermore, we describe the importance of co-creation better. Two paragraphs were added (please see the new paragraphs in the response to reviewer 1) or within the manuscript (page 2, line 85-92 & page 3, line 115-124).

Round 2
Reviewer 1 Report
Authors have completely addressed all my concerns.